# Nord Stream methane leaks spread across 14% of Baltic waters

Martin Mohrmann [1,2] ✉, Louise C. Biddle [1,2], Gregor Rehder [3], Henry C. Bittig [3] & Bastien Y. Queste [2]

A suspected 443-486 kt of methane escaped from the Nord Stream pipelines in September 2022 at four explosion sites across three pipelines. Much of this methane rapidly escaped to the atmosphere, while an unknown amount was dissolved. We use sustained high-resolution observations of methane concentrations from autonomous gliders and an instrumented ship of opportunity to reveal the timing and spread of dissolved methane across different Baltic regions and marine protected areas. Estimates of methane spread and concentrations are essential to understand the ecosystem response, and for establishing accurate priors for atmospheric outgassing and transport models. A numerical model, initialized by engineering estimates and our observations, enables us to constrain the mass of locally dissolved Nord Stream methane (9.5-14.7 kt). We show that dissolved methane decreased rapidly through outgassing, however initial concentrations were so high that 14% of the Baltic Sea still experienced concentrations 5 times greater than average natural levels.

Nord Stream 1 and 2 are two double-strand natural gas pipelines that run submerged at the bottom of the Baltic Sea from Russia to Germany. On September 26, 2022 four explosions damaged the pipelines, and in consequence, three pipelines started to leak in the Bornholm Basin. At the time of the incident, all pipelines were pressurized with natural gas (methane content >96%[1]), but not used for transport. The gas leaks were observed as a sudden drop in pressure by the Nord Stream station Lubmin, by a seismic signal of underwater explosions at the seafloor of the Bornholm Basin, and successively from planes and satellites that recorded large (~1 km) surface plume expressions over the leak sites. The surface bubbling ceased by October 2, 2022 and in total 443–486 kt of methane had escaped into the Baltic Sea[1], of which the largest part escaped into the atmosphere after rising to the surface. It was the largest single-event methane emission recorded in recent times.

In the environmental assessment report commissioned by the pipeline operator Nord Stream AG, the mass of methane with the possibility to escape was estimated to 148 kt from a single pipeline rupture[2], which is less than the estimate in ref. 1. Moreover, the possibility of multiple pipeline ruptures, in this case three at the same time, was neglected and thus the actual total methane release was about three times greater than the worst case estimate (148 kt) included in the report. Moreover, the solubility of methane in the seawater was considered to be negligible, assuming all methane would rise to the surface and be released into the atmosphere, with little effect on water quality. Although the solubility of methane is relatively small, a substantial part of the escaped methane dissolved into the seawater (section "Model results of methane spread and fate").

Atmospheric methane is a major driver of global warming[3]. A water mass is commonly called methane-saturated if it is in equilibrium with the atmospheric partial gas pressure of ~1 atm × 1900 ppb in 2022[4]. The Bornholm Basin is commonly supersaturated by 10–60%. Sources of methane include the anoxic microbial degradation of organic matter in the sediments as well as lateral transport of enriched waters from the neighboring Arkona basin[5]. During the Nord Stream leaks, methane was entering the water at a depth (~70 m) where high

[1]Voice of the Ocean Foundation, Skeppet Ärans Väg 19, Västra Frölunda 426 71, Sweden. [2]Department of Marine Science, University of Gothenburg, Box 463, Göteborg 405 30, Sweden. [3]Leibniz-Institute for Baltic Sea Research, Seestrasse 15, Rostock D-18119, Germany. ✉ e-mail: martin.mohrmann@voiceoftheocean.org

pressure and low temperature increase methane solubility. Methane is non-toxic at levels commonly encountered in the environment, but the extreme concentrations of the Nord Stream leak release could pose unpredictable sublethal risks to the ecosystem, while also affecting biogeochemical pathways[6].

In this study, we present continuous high-resolution in situ observations of dissolved methane from a glider and a ship of opportunity (SOOP) traversing between Lübeck and Helsinki, spanning over a duration of 3 months after the Nord Stream pipeline ruptures. The glider sampled the whole water column a few km downstream of the leaks, starting 9 days after gas started to leak from the pipelines, whilst the SOOP observations provide a larger spatial view of methane concentrations in the surface layer, and were already ongoing prior to the incident.

To extend our analysis spatially to the whole Baltic Sea and temporally to the hours/days immediately following the Nord Stream incidents, we use the Lagrangian chemical-fate and transport model OpenDrift/ChemicalDrift[7,8]. The model has been designed for modeling the trajectories and fate of substances drifting in the ocean, and has a track record in the simulation of volatile dissolved substances[9] and oil spills[10]. The numerical model is calibrated to our observations to simulate the advection, diffusion, and outgassing of the methane. Overall, we provide methane concentrations over time, depth, and space, both collected in situ and completed with numerical model data to reveal the magnitude and persistence of Nord Stream methane across the Baltic Sea.

## Results

### High-resolution in situ CH4 measurements

We measured methane concentrations continuously (Fig. 1) for the 3-month period after the leaks. The glider and the SOOP transited back and forth along the transects marked by white and red lines in Fig. 1a, passing the three northern leak sites at a distance of ~9 and ~20 km, respectively. The SOOP system, operated as part of the European ICOS Research Infrastructure, is ideally suited to monitor surface methane concentrations across the Baltic Sea long term and provides the longitudinal spread of the dissolved methane concentration plume

across the surface, while the gliders provide a unique high-resolution view of the plume's vertical structure in the water column.

We consider 10 nM as a natural and average background methane surface concentration. The natural surface layer concentration is often lower in autumn (5 nM, refs. 5,11) but at depth (halocline depth ~50 m) it can be up to 50 nM[5]. Increased methane concentrations (>10 nM) in the surface layer of up to 385 nM ($c_{max}$) were first recorded by the SOOP on September 29, 2022 over a length of 30.1 km ($l_{c>10nM}$) of the ship's trajectory. However, during the subsequent passages over the leak sites on October 1, the signal had mostly disappeared (<10 nM), probably due to advection. High methane concentrations were observed next on October 4 ($c_{max}$ = 220 nM, $l_{c>10nM}$ = 39.7 km) and became gradually stronger and more widespread on October 5 ($c_{max}$ = 1850 nM, $l_{c>10nM}$ = 133.8 km), October 7 ($c_{max}$ = 1475 nM, $l_{c>10nM}$ = 128.2 km), and reached a peak in concentration on October 8 ($c_{max}$ = 3070 nM, $l_{c>10nM}$ = 153.5 km, Fig. 1b). On October 5, 9 days after the leaks were discovered, we deployed an ocean glider with a methane sensor at a distance of 24 km from the leak sites (Fig. 1a). On the same day, as the glider moved as close to the leak sites as the exclusion zone allowed (~9 km distance), we observed a sharp increase of methane concentration up to 10,000 nM; values two to three orders of magnitude higher than typical background concentrations [refs. 5,12, note that measurements above the 1000 nM upper limit of the methane sensor's calibration are potentially less accurate]. However, 2 days later, as the glider traveled to the further, south-eastern end of the transect from the leak sites, methane concentrations dropped to ~10 nM in the water column (Fig. 1a–d), indicating the plume had not yet spread that far. This strong spatial gradient diminished rapidly over a duration of 5–10 days, as methane concentrations remained high (~1000 nM) across the whole transect for another 2 weeks, before eventually showing a gradual decrease (Fig. 1b–d).

By November 1, over a month post-leakage, column-averaged methane concentrations remained ten times higher than typical background concentrations (Fig. 2a). Despite the glider and SOOP transects not spatially overlapping, the magnitude and rate of decrease of the concentrations are consistent between both platforms

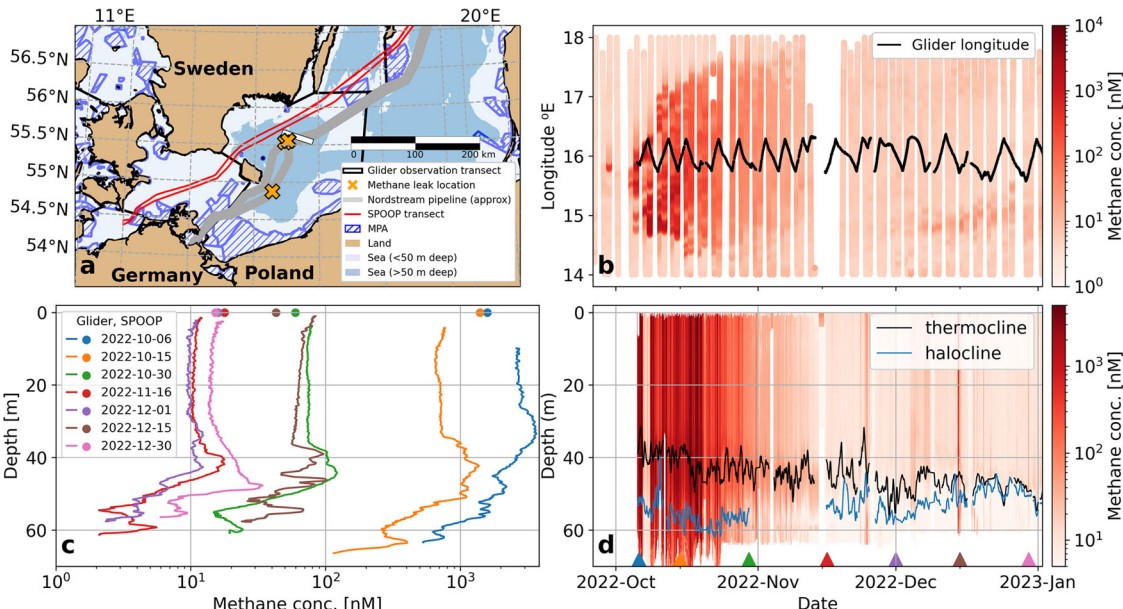

**Fig. 1 | Three-month in situ observations of methane concentrations in the Baltic Sea after the Nord Stream explosions. a** Map of the southern Baltic Sea with the Nord Stream pipelines (gray), the leak locations (orange crosses), glider sampling trajectory (white line), the trajectory of the SOOP (red lines), and Marine Protected Areas (dark blue). **b** Hovmöller diagram of SOOP-observed surface methane concentrations (red colors) with a longitudinal position of a glider (black line). **c** Coincident individual profiles of glider-observed methane concentrations (lines) and nearest SOOP surface observations (dots). **d** Glider observations of methane concentration throughout the water column. All methane data are presented on a logarithmic scale.

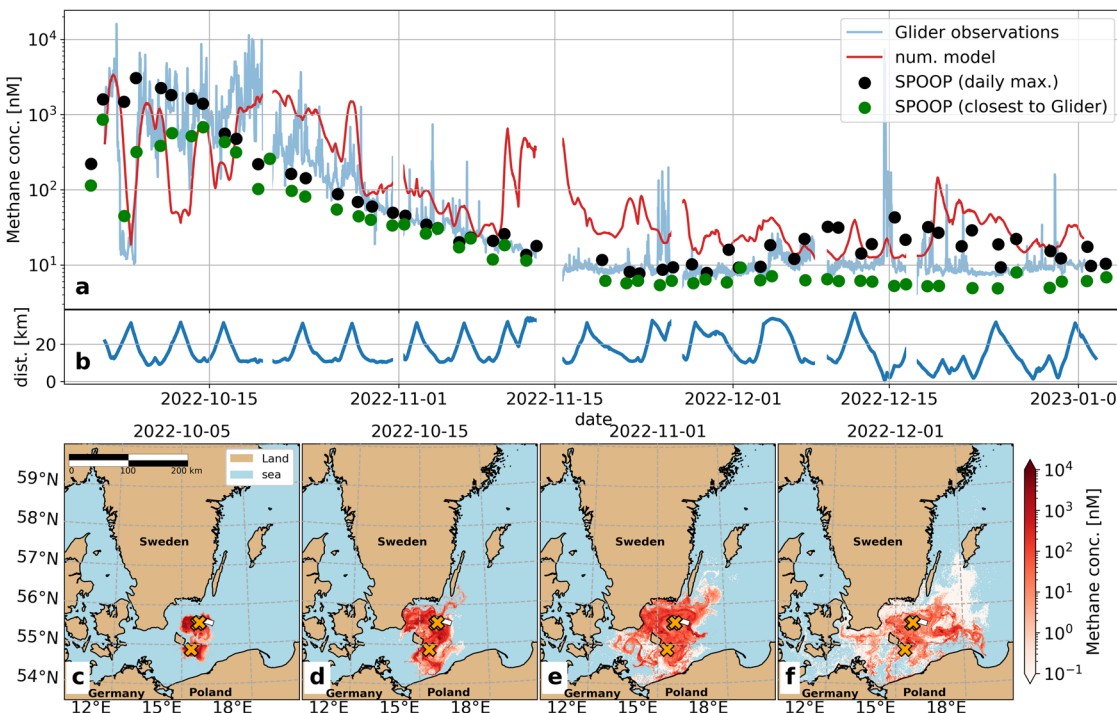

**Fig. 2 | Model results of dissolved methane concentrations.** All concentrations are depth-averaged and presented on a logarithmic scale. **a** Glider observed (blue) and modeled (red) methane concentration over time. **b** Distance of the observations/subsampling in km from the northern leak sites. **c–f** Snapshots of methane spread from numerical simulation.

(Figs. 1c, 2a). By mid-November, concentrations had decreased to levels indiscernible from typical background concentrations. Following this period, most glider observations show near-background concentrations of methane. However, isolated patches of high dissolved methane concentrations were observed more than 2 months after the leaks (>1000 nM on December 14, Figs. 1d, 2a) in direct proximity to the leak sites (~1 km, Fig. 2b) when the exclusion zone was lifted. Localized patches of high methane near the leak site were present until the end of our survey on January 2, 2023, indicating some kind of sporadic gas discharge from the site.

Glider observations show that the vertical methane plume structure is governed by oceanographic conditions. In this region and at this time of year, the water column consists of three stratified layers (Fig. S1). The surface layer, in contact with the atmosphere, is warm and fresh in autumn, and extends down to the seasonal thermocline (30–40 m; Fig. 1d), which is created by solar heating from the surface. The bottom of the water column consists of higher salinity water which flows in intermittently from the North Sea, showing intermediate temperatures, higher salinity, and very low oxygen concentrations due to its isolation from the atmosphere by strong stratification (the halocline, Fig. 1d). In between is an intermediate layer of cold fresh water left over from winter conditions when the seasonal thermocline coincided with the deep halocline. This intermediate layer is entrained into the surface layer as the surface cools and storms mix in autumn.

At all times, we observed the highest methane concentrations at mid-depth between the thermocline and the halocline (Fig. 1d). While the depth of the halocline stayed at ~50 m throughout the survey, the surface mixed layer, with an initial depth of ~30 m in late September, deepened to ~40 m by the second half of November, and eventually eroded to full halocline depth by mid-December (Fig. S1). As a consequence, waters from the former winter mixed-layer with the strongest enrichment in methane (Fig. 1d) were successively entrained into the surface layer for several weeks after the accident. The observed depth structure of methane concentration supports the findings of

ref. 13, who observed the highest methane concentration (albeit caused by a gas blowout in the North Sea) around the pycnocline, due to the deflection of the methane-enriched plume by the density gradient.

As the surface mixed layer deepens during autumn, this intermediate layer becomes thinner (Fig. 1c, d and Fig. S1). Lowest methane concentrations were consistently observed in the bottom water below the halocline (>50 m). The surface layer showed intermediate concentrations that were lower than the winter water and higher than the bottom water. Despite a clear and consistent vertical pattern, the gradients observed are predominantly in the temporal/horizontal dimension. This can be explained by an initially homogeneous vertical distribution of methane driven by the rapid and strong agitation of bubbles floating through the water column, and the turbulent mixing they caused, causing rapid dissolution of methane. As Baltic circulation is dominated by barotropic (i.e., depth-uniform) flow, the observed vertical structure will not vary as much as the horizontal or temporal patterns. The relatively weak depth-dependency also implies that the SOOP observations can explain much of the pattern, especially in winter when the mixed layer is deep, with implications for long-term monitoring or future methane blowouts.

## Model results of methane spread and fate

The glider and SOOP observations provide a unique dataset to understand local, horizontal, and vertical processes that governed methane concentrations within the Baltic Sea after the leakage. We used these data to tune the initial conditions of a Lagrangian chemical-fate and drift model (Section "Model") to simulate advection, diffusion, and outgassing and to map the spread and residence time of methane across the wider Baltic Sea.

The bacterial oxidation of methane is neglected here because (1) the oxidation rate is multiple times smaller than the concentration decrease due to outgassing and dilution, at least during the first month of simulation (see Section S3), and (2) there are large uncertainties concerning oxidation rates in the water column and even in the

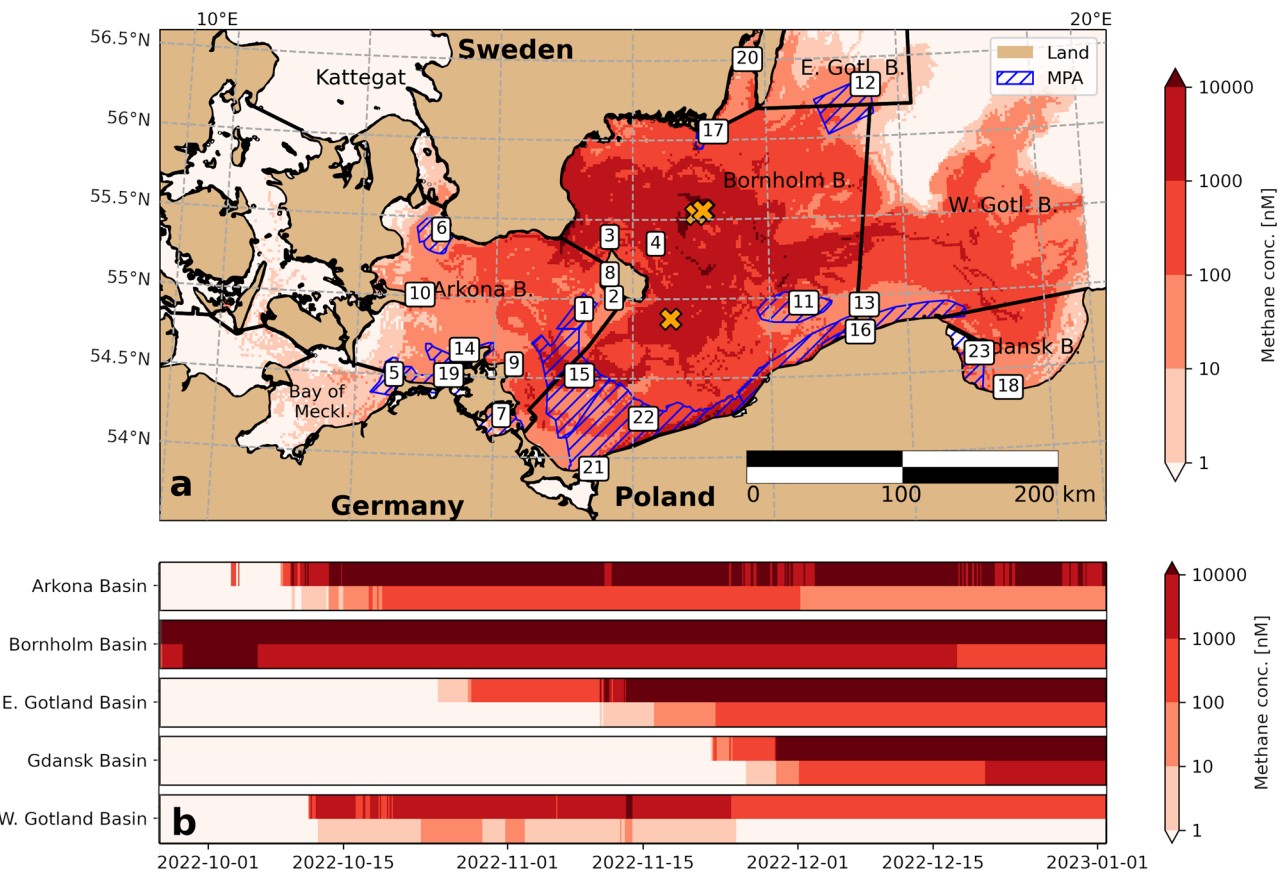

**Fig. 3 | Methane concentration exposures. a** Map of maximum methane exposure from Sept 2022 to Jan 2023, the number labels refer to the underlying MPAs, listed in Fig. S5. **b** Basin-specific maximum (upper part of the bar) and mean (lower part of the bar) methane concentrations.

abundance of large amounts of dissolved methane the oxidation rate can change drastically over time[6]. However, we provide a sensitivity analysis of oxidation rates in Supplementary Section S3.

By iterative linear regression between the modeled and observed concentrations for the first day of the glider deployment (Supplementary Fig. S2), we estimate that a total of 10.8 kt of methane dissolved into the water, which equals initial concentrations at the leak sites of 230,000 nM. (Supplementary Section S2). This estimate is very similar to an estimate independently derived by ref. 14 of 11 kt based on numerical modeling.

An amount of 10.8 kt is our best estimate, because early model data have the smallest accumulation of potential errors in the advection, diffusion, and outgassing modeling. We used output from the CATHARE rupture model for this best estimate because it has lower errors in the cumulative pipeline methane content compared to PREAK[1]. To control the robustness of the estimate and to derive an uncertainty measure, we compare our model input also with the entire timeline of glider data, and with the SOOP surface data. A tuning with all glider observations results in an estimate of 7% more methane (11.6 kt), tuning with the SOOP data results in 21% more methane (13.1 kt), and the use of the alternative pipeline rupture emissions modeling presented in ref. 1 (PBREAK) resulted in 3% less methane (10.5 kt). The uncertainty estimate for the linear regression plot slopes is about 10%. The small spread of these independent estimates supports the robustness of our findings. In summary, the likely range of the total amount of Nord Stream methane that dissolved in the seawater is between 9.5-14.7 kt. This number only accounts for ~3% of the total amount of gas released into the atmosphere[1]. Still, the fraction of methane that was dissolved is large compared to the natural existing dissolved methane pool, and controls additional emission to the atmosphere several weeks after the incident.

Our model replicates the observed gradients during the first 2 weeks after the leaks and the anti-exponential decrease from mid-October to mid-November (red and blue line in Fig. 2a). After November, patches of increased methane concentrations remain, especially at depth. After the first week of glider observations (2 weeks after the leak), the modeled methane concentrations deviate from the observations, because errors of individual Lagrangian parcel simulations accumulate over time. However, the main usage of the model results is to inspect the fate of the methane and the extent of its spread over time. These characteristics are robustly represented by the statistical averaging of hundreds of thousands of Lagrangian parcels. The agreement between the model and observation indicates that (a) we identified the first-order processes as advection/diffusion and outgassing correctly for the weeks after the leak and (b) we can constrain the rate of microbial oxidation to the difference between model and observations.

Outgassing rates are, on average, 8.6 t h$^{-1}$ for October, 2.6 t h$^{-1}$ for November, and 0.6 t h$^{-1}$ for December, and short-term variability is mainly governed by wind speed. The maximum outgassing rate occurs on 6 October at 55.1 t h$^{-1}$. Note that we only consider the outgassing of dissolved methane; for the transport by bubbles from the leak to the surface, we refer instead to related papers[1,14,15]. Outgassing was observed directly with an airborne instrument[16], and was estimated to be between 19 and 48 t h$^{-1}$ on October 5, our model estimations are (8–20 t h$^{-1}$) for the same day. The confidence intervals overlap, the difference in the central estimates could be caused, for example, by the (initial) depth distribution of the methane in the water column[16] or an inaccurate transport direction of the methane due to errors in the underlying oceanographic model, which are uncertainties in our model tuning (see Section S2 and "Model results of methane spread and fate").

According to our model simulations, of the 16 subbasins of the Baltic Sea[17], 10 Basins experienced clearly increased methane concentration (more than twice natural concentrations). These basins are, in alphabetical order, the Arkona Basin, Bay of Mecklenburg, Bornholm Basin, Great Belt, Eastern and Western Gotland Basins, Gdansk Basin, Kattegat, Kiel Bay and The Sound (Fig. 3). Out of these ten affected subbasins, three experienced at least tenfold increased methane concentrations (Kiel Bay, Great Belt, Bay of Mecklenburg) and another five experienced at least hundredfold increased methane concentrations (Fig. 3b; Arkona Basin, Bornholm Basin, Eastern/Wester Gotland Basins, Gdansk Basin). Of 188 Marine Protected Areas (MPAs) in the Baltic Sea[17], eight were exposed to methane concentrations 100 times higher than the norm, and another ten and five MPAs to at least tenfold and twofold increased methane concentrations, respectively (see Supplementary Fig. 4). The model shows that 14% of the Baltic Sea area experienced concentrations five times above background levels (>50 nM).

The major environmental impact of the leaks is the emission of methane to the atmosphere[1,18] where it acts as an atmospheric greenhouse gas contributing to global warming. Multiple stations of the integrated carbon observation system detected methane concentrations well above natural variability[19], but the plume dispersed within some days, and on October 5, the atmospheric concentrations in the vicinity of the leaks (~10 km) were within 20% of the background concentration, with the remaining concentration caused by outgassing from the sea[16]. In contrast, the observed concentration of dissolved methane in the Bornholm Basin increased by two orders of magnitude for at least three weeks (Figs. 2, 3). Longer residence time in the water buffered outgassing, leading to prolonged but reduced supply to the atmosphere. The regional bottom waters (sub-halocline waters) are known to be temporally severely oxygen deficient and enriched in methane[5], supporting the presence of methane-oxidizing microorganisms. In the transition zone between oxygenated and suboxic waters (redoxcline), these microorganisms have been shown to quickly adapt to changes in substrate availability[20], so they might have been temporarily enriched, but no long-term shifts are expected. Studies on microbial community response are ongoing (K. Abrahamsson, pers. comm.), but there are no publications to date. Due to the minor fraction of non-methane components in the Russian natural gas[1], the effects of other components have not been reported. However, the immediate shock waves of the NS explosions are suspected to affect marine mammal populations, including the critically endangered harbor porpoise, while resuspension of contaminated sediments creates immediate risks for fish and other biota[18] and might also encourage further oxygen drawdown from the water column and affect benthic communities. Results of the environmental monitoring program in the Baltic Sea for the year after the accident (nutrient distribution, phyto- and zooplankton monitoring, accessible in the IOW ODIN database) did not indicate unusual deviations from the long-term observations and trends. This supports early statements suggesting minor impacts on the marine environment and ecosystem health.

## Discussion

Industrial accidents, extreme weather events, or infrastructure disruptions require response times of hours to days to study their immediate impact, but they can be hard or impossible to predict. The observed exponential concentration decrease over a matter of days/weeks stresses this urgency. The unique availability of gliders, methane sensors, and existing monitoring programs (the SOOP Finnmaid and VOTO observatories) coincided to allow immediate monitoring and maintain high-resolution observations over 3 months without disturbing ongoing international investigations. This demonstrates the strength and the potential for uncrewed systems such as gliders and SOOP. Our observations provide the most robust estimates to date of resulting ocean methane concentrations in the Baltic Sea after the Nord Stream accident, through the combination of two observational datasets (horizontal and temporal through the SOOP, vertical and temporal through the gliders), which create the possibility to accurately model spread and residence time. Even more valuable is the availability of the marine dataset to serve as a prior to refine atmospheric outgassing and transport models presented in ref. 1.

The leaks released unprecedented amounts of natural gas into the water and the atmosphere, making a clear prognosis of ecosystem effects difficult. The impact also depends on the probability or frequency of ruptures, that was estimated to be one event in 20,000 years in the Nord Stream environmental impact assessment[2]. The rupture of the Nord Stream pipelines in 2022 and the Balticconnector pipeline between Finland and Estonia only 3 years after commissioning[21], requires that the impact assessments for subsea pipelines must be adapted to the current geopolitical context. However, this study provides a robust record to understand potential sublethal impacts on the ecosystem that may be discovered in the future, identifying 23 MPAs that experienced significantly heightened levels of methane. Apart from that, our study shows that the majority of the initially dissolved methane eventually outgassed to the atmosphere, where it increased the GHG emissions for Europe.

## Methods

### Glider observations

In situ methane concentrations were collected by sequential SeaExplorer glider[22] deployments between October 5 2022 and January 2 2023. All gliders were equipped with the same methane sensor, a Franatech METS methane sensor. Gliders were either equipped with an RBR Legato CTD and Coda oxygen sensor, or a Seabird GPCTD and SBE43F oxygen sensor. Sensors recorded at 1 Hz, equivalent to ~10 cm vertical resolution throughout the mission.

The Franatech METS sensor was mounted on the outside of the glider in all configurations (both pumped and unpumped CTD). Flow past the sensor was ensured by the glider's motion through water (~35–40 cm s$^{-1}$). The sensor used a heated tin-dioxide-based semiconductor which detects the partial pressure of combustible gas such as methane[23]. This semiconductor is located in the sensor behind a semi-permeable silicone membrane which separates the aqueous phase of the external environment and the gaseous phase in the sensor. Multi-point temperature and methane calibrations for the 10–1000 nM range were performed by the manufacturer before and after the mission. An additional calibration to determine oxygen sensitivity was performed after the mission.

Several factors complicate the use of the Franatech METS sensor in this environment. The gliders are constantly in motion, which gives the sensors limited time to adapt to changes. However, the diffusion of methane across the membrane of the sensor is not instantaneous. A relatively slow response time ($\tau_{methane}$) of the sensor could lead to an underestimation of peak values and a hysteresis curve, which must be corrected. Second, we need accurate temperature of the gas phase to calculate methane concentration from the partial pressure; water temperature is known accurately from the outside, and the semiconductor temperature is known through the sensor. A second lag ($\tau_{temperature}$) defines how quickly the internal gas phase temperature reacts to changes in temperature. Third, the sensor response is oxygen-dependent. The post-mission calibration included the determination of sensor response under decreasing oxygen concentrations to account for near-anoxic conditions in the bottom waters of Bornholm Basin; the correction is a nonlinear scale factor ranging from ~0 and 1 corresponding to anoxic and normoxic conditions, respectively. A third lag ($\tau_{oxygen}$) is used to estimate oxygen concentrations within the gas chamber of the sensor. Finally, the presence of hydrogen sulfide is suspected to bias the sensor response.

## Table 1 | Basic model equations

| Component | Description | Input Parameters |
|---|---|---|
| Advection | Euler method[7] | $u$, $v$, $w$[29] |
| Diffusion | Random walk displ. loop[7] | $D_{h,MLD} = 5.0$ m²s⁻¹ [36], $D_{v,MLD}(\tau_{wind})$[37] |
| Outgassing | Two film[8] | $T$, $S$, MLD, wind[29,30] |
|  |  | $H^{cp} = 1.4 \times 10^{-5}$ mol m⁻³ Pa⁻¹ [33] |
|  |  | d(ln $H$) /d(1/$T$) = 1900 K[33] |
| Oxidation* | First-order degradation[8] | $k = [0.035, 0.0175, 0.0035...]$ d⁻¹ [14] |

Variable names in the table are u, v, w for the horizontal and vertical current velocities, the diffusion coefficient D, the temperature T, salinity S, mixed layer depth MLD, Henry's law constant $H^{cp}$, and its temperature dependency d(ln H)/d(1/T), and the first-order oxidation constants k. *Oxidation was only applied in supplementary information, k = 0 in the main text.

We utilize the unique characteristic of glider profiling to empirically regress the relevant sensor lags and methane concentrations as the glider profiles through methane, oxygen, and temperature gradients. We empirically regressed the lag coefficients using a Nelder-Mead simplex algorithm and scoring function which minimized the area between successive up and down casts in methane-pressure space, methane-temperature space and methane-oxygen space. The same method is routinely applied to regress lags for other sensors on gliders[24]. Optimal results were obtained with a $\tau_{methane}$ of 134 s, $\tau_{temperature}$ of 3 s, and a $\tau_{oxygen}$ of −24 s. Data were discarded after diving through hydrogen sulfide (visible through anomalous values in the SBE43F) during the initial month, and then dives were cut at a shallower depth for the rest of the mission to avoid contamination. As such, we do not have methane values near the seabed for most of the mission. The calibrated, lag-corrected, and oxygen-corrected methane data agree very well with the data collected by the IOW system on SOOP Finnmaid (Fig. 1).

## SOOP surface CH4 observations

Surface methane concentrations were continuously recorded on board the SOOP Finnmaid, traversing continuously between Travemünde (Germany) and Helsinki (Finland) with a round-trip time of 3 days. The instrumentation on board is a German component of the European Integrated Carbon Observation System Research Infrastructure. The partial pressure of methane has been continuously recorded on the vessel since 2010, and the system is described in detail in refs. 25,26. Water is pumped continuously from  -3m water depth into the ship, and the water is equilibrated with a recirculating airstream using a hybrid of a shower head and bubble type equilibration system. The recirculating air is partially dried and the methane (and carbon dioxide) mole fractions are measured by means of an off-axis integrated cavity output spectrometer (oa-ICOS; Los Gatos Research CH₄/CO₂ gas analyser). Sensor calibration is performed every time the ship is in the harbor (about every 30 h) using three reference gases provided by the ICOS Central Calibration lab. High-accuracy water temperature sensors at the water inlet and within the equilibration vessel allow for correction of warming, which is usually held well below 0.5 °C. With these data, salinity, and total pressure, data were converted to methane concentration using the Bunsen solubility given in ref. 27. With a recording frequency of 1 min, the spatial resolution is about 700 m (at a ship's speed of  -23 kn). For further details, we refer to refs. 25,26. The equilibration system differs from the description in the referenced literature due to a major upgrade of the system in 2019, where an improved equilibration system of larger volume, as well as additional instrumentation, was amended. The overall accuracy of the methane measurements is usually better than 1%, though the high concentrations encountered in the first days after the Nord Stream accident fall out of the calibration range.

However, a larger uncertainty of the data relates to the slow response time of the methane equilibration process, usually about 10 min, which causes a spatially biased smoothing of the signal. Due to the steep gradients in surface methane concentrations, it was necessary to implement a time-lapse correction (see refs. 28,26, their Appendix A2). During the observation period, the water flow to the system gradually decreased, but we refrained from a maintenance break to assure undisrupted data coverage. Several routines were tried to achieve the best consistency of data, which can be assessed by a match of the general patterns of the East-West and West-East transects. In the end, the response time was parameterized as a function of the water flow into the equilibration system, which is also recorded with a temporal resolution of 1 min. Time constants for methane ranged from typically 15 min at the nominal flow rate to a maximum of 60 min towards the end of the survey in January.

## Model

A lagrangian model called ChemicalDrift[8] based on OpenDrift[7] was used for our numerical spread simulations. The model's input data consisted of ocean currents, bathymetry, mixed layer depth and grid specification for the advection/diffusion modeling; and salinity, temperature, mixed layer depth and wind speed for the outgassing component. The Baltic Sea Physics and Analysis Forecast Model (BAL-MFC;[29]) is used for all input data except for the wind speed, here we use ERA5[30]. The models have a spatial resolution of 0.028 × 0.016° with 56 depth levels (BAL-MFC) and 0.25 × 0.25°, respectively, with a timestep of 1 h. Our Lagrangian model does not have a spatial model resolution in a strict definition but is limited by the resolution of the forcing data.

Our model starts with the first leak on September 26, 2022 and ends with the end of our in situ glider observations on January 2, 2023. We continuously initialize methane tracer in a radius of 5 km from the leak site, the relative amount of methane initialized is a constant fraction of the engineering estimates of total pipeline methane emissions (CATHARE, PBREAK[1,31,32]). Our model domain extends from 10 °E to 20 °E and from 53.5 °N to 60 °N. From the leak sites, the methane is transported by the currents according to the Euler advection equation

$$P^{\mathbf{X}}_{t+\Delta t} = P^{\mathbf{X}}_t + u(\mathbf{X}, \tau)\Delta t \qquad (1)$$

with the position of the Lagrangian body $P^{\mathbf{X}}_t$ at the time step t (timestep length $\Delta t$), the seawater velocity $u(\mathbf{X}, \tau)$. In the Lagrangian formulation, the diffusion is expressed as a random walk displacement loop

$$P^{\mathbf{X}}_{t+\Delta t} = P^{\mathbf{X}}_t + \int_t^{t+\Delta t} u(\mathbf{X}, \tau) + u_{noise} d\tau \qquad (2)$$

with

$$u_{noise} = \sqrt{\frac{2D}{\Delta t}} R_{h,v}(z) \qquad (3)$$

where $R_{h,v}(z)$ are random numbers taken from a normal distribution with zero mean. The dissolved methane concentrations are modeled as a passive tracer in the advection/diffusion equations.

$$\frac{\partial}{\partial t} C_{water} = -\frac{MTC_{vol}}{H_{MLD}}(C_{sat} - C_{water}) \qquad (4)$$

The implementation assumes a negligible methane concentration in the air for simplicity. In our model, we can justify this simplification because (1) the initial methane concentrations are more than three orders of magnitude higher than a typical background concentration of methane in equilibrium with the atmosphere and (2) we are interested only in the additional methane concentrations caused by the Nord Stream leaks. A typical constant methane background level of

10 nM is added after the simulation. Instead of using the approximation method provided in ChemicalDrift, we chose to specify Henry's volatility (outgassing) constant Hpc manually, using the value provided in ref. 33. Oxidation (used for sensitivity study in supplementary information only) is modeled using a first-order degradation equation

$$\frac{\partial}{\partial t} C_{water} = C_{water}\left[1 - \exp(-k\Delta t)\right] \tag{5}$$

Input parameters are listed in Table 1. Further parameters and implementation/discretisation details can be found in the the provided source codes[34].

At the end of the simulation, we grid the Lagrangian model output to the same resolution as the original (BAL-MFC) ocean model resolution. Then we add the mass of all lagrangian particles in the grid cell and divide it by the grid cell's volume to compute concentrations, which can be compared to the observations. The results for concentration are most robust for grid cells with a large amount of lagrangian particles and lose precision if only a few particles are present in a grid cell. However, for our estimations of areas with hundredfold, tenfold, or twofold increased methane concentrations, we have averages of 72, 53, and 41 particles in each horizontal bin on average. The nonlinear dependency between concentration and particle number is due to differences in particle mass, affected by outgassing and grid cell volumes.

To evaluate which marine protected areas (MPAs) and subbasins are affected, we use the HELCOM area specifications[17] and ref. 35. We reduce the four-dimensional model output data (latitude, longitude, depth, time) by aggregating first over the depth coordinate using a column mean and then over the time coordinate, choosing the maximum value to estimate a two-dimensional maximum exposure for all horizontal grid cells. Now, either (1) threshold filters are applied to filter for all (horizontal) grid cells above specific mean concentrations, enabling us to estimate the total polluted area. This area can then be compared with the area of the entire Baltic Sea (sum of all subbasin areas from ref. 17). Or alternatively (2) the grid cells are binned into the individual subbasins for Fig. 3 or MPAs for S5 respectively. Both, the maximum and mean methane concentrations within each individual basin/MPA are presented in Figs. 3 and S5 to account for spatial gradients within these relatively large areas.

## Data availability
The glider observations used in this study are available in an ERDDAP database at observations.voiceoftheocean.org. The dataset identifiers on the server are SEA070_M13, SEA070_M14, SEA070_M15, SEA056_M54, SEA056_M55, SEA056_M56, and SEA056_M57. Furthermore, we used datasets SEA077_M11, SEA069_M11, SEA077_M13, SEA045_M69, SEA077_M15, SEA045_M71, SEA077_M17, and SEA045_M73 to describe regional oceanographic conditions in section "High-resolution in situ CH4 measurements" and Fig. S1. The DE-SOOP Finnmaid data used in this study is available via the ICOS Carbon Portal, https://doi.org/10.18160/K3BM-8YNG. IOW Baltic Sea monitoring data were available at https://odin2.io-warnemuende.de/.

## Code availability
The codes used to analyse the data and create the plots are provided in ref. 34.

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

## Acknowledgements
We thank Raphaël Préa for providing CATHARE pipeline emission estimates for methane, that we used in our numerical model. Jens Gronnema and Michel Masson of Franatech for advice on improving the quality of the Franatech METS sensor data. Alseamar for offering to make one of their Franatech METS sensor available at such short notice. Michael Glockzin for skillful work on the SOOP Finnmaid data. The German Federal Ministries for Education and Research (BMBF) and Digital and Transport (BMDV) for support of ICOS Germany. Nicolai von Oppeln-Bronikowski and one anonymous reviewer for their helpful comments. The VOTO operations team for deploying gliders frequently, at short notice. B.Y.Q.'s time was funded by the Voice of the Ocean Foundation, the EU GROOM II grant (ID: 951842), and the Swedish Formas grant 2022-01536. B.Y.Q., L.C.B., and M.M. thank Alseamar for a loan of the Franatech METS sensor and Jens Gronemann at Franatech for support in processing the methane data.

## Author contributions
M.M. drafted the paper, produced all figures, and set up and performed all particle tracking model runs and calculations. L.C.B. contributed to drafting the paper and to the development of figures. B.Y.Q. was the PI of the project and led the analysis of the glider data and developed the correction scheme to process methane sensor lags. B.Y.Q. co-wrote the paper and contributed to the development of figures. G.R. and H.C.B. provided the SOOP data. H.C.B. did the time-lapse correction of the SOOP data. G.R. contributed to the writing.

## Competing interests
The authors declare no competing interests.
