## [Peer Review File · Nature Communications]

Nord Stream methane leaks spread across 14% of Baltic watersREVIEWER COMMENTS

Reviewer #1 (Remarks to the Author):

Review of “Nord Stream methane leaks spread across 14 of Baltic Waters”

This study outlines the measurements and mapping of the Baltic Sea surface and subsurface signature elevated CH₄ levels caused by the Nord Stream 2 natural gas pipeline leaks. The authors describe the curation and processing of autonomous glider and vessel-based sea-surface methane concentration measurements with agreement between platforms. These measurements and a lagrangian tracer analysis tool (ChemDrift) fed by a regional ocean forecast system are used to predict the subsurface spread of the methane retained in the water column from the leak sites. Their spread and concentrations in several Baltic Sea Marine Protected Areas (MPAs) are discussed in the text. The authors arrive at an estimate of ~10.9kt dissolved methane from the leak, or roughly 2.5% of the total emitted methane that escaped to the atmosphere (445 kt). The dissolved methane was predominantly stored in the intermediate subsurface layer above a local halocline and below the seasonal mixed layer, allowing it to spread through currents. The authors report increased dissolved CH₄ levels in 23 MPAs by factors ranging from 2 to 100 depending on the distance to the leak site. Some dissolved methane was later entrained into the winter-time mixed layer and outgassed to the atmosphere. Overall, the primary societal impact associated with increased dissolved methane levels in the Baltic from the leak appears to be the additional contribution to Europe’s GHG emissions. The research demonstrates the value of rapid marine environmental assessments in a unique socio-political and economic landscape and physically dynamic environment. The benefits of underwater gliders and other marine observing infrastructure to society are evident.

The paper is clear, well-written, and of broad interest to many disciplines and the general public. The information presented supports the overall findings as presented by the authors. The methodology is clear, but some passages in the text could be improved for clarity. The conclusions arrived at in the study could be strengthened by comparing the results to those anticipated in the initial impact assessment for the Nord Stream 2 project. The long-term impact of the leak on the marine environment near and away from the leak site is also unclear. More context should be provided on how this event impacts the surrounding marine ecosystem. Additional specific comments/suggestions are included below. I think the paper should be accepted with minor revisions.

Thank you for an enjoyable paper to read and review.

General comments:

One aspect I think should be included is how this event compares with the original impact

assessment report done by Nord Stream AG, “Chapter 9: Assessment of risks related to unplanned events”. The report mentions the worst-case release of 148kt during a pipeline leak. That is ~3.7 times lower than the worst case that occurred (number in the text 445 +/- 25 kt). The impact assessment also does not discuss any marine impact due to the low solubility of methane. Given your findings, how would you characterize this report? It would be good to include this original assessment in the introduction and compare the expectations with what happened so that in the future, this type of knowledge could be integrated into future projects of this nature. A few sentences in the introduction/discussion/conclusion section and abstract would enhance this paper’s findings.

The authors note that the main environmental impact of the pipeline leak will be elevated GHG emissions. What is the broader marine impact of this event, if any? Some additional comments in the conclusion/introduction could relate this question to the findings.

Specific comments:

Correction of the glider CH4 measurements: The corrections are complicated with three distinct dependencies ($\tau_{\text{ch4}} + \tau_{\text{temperature}} + \tau_{\text{oxy}}$). I was looking for this correction process online in the GitHub link to the paper review, and I could not find this information. I suggest sharing a code example with data to benefit others in the research community who may want to use this system on either gliders or floats. It appears to be a promising solution.

Glider CH4 sensor integration: It was not mentioned if the CH4 sensor was set up with a pumped flow head to increase gas flow through the membrane.

L83-84 - Consider mentioning how that calibration was done in the appendix.

L86-87 - sentence unclear/ hard to understand.

L110 - How well did the data agree with each other? What was the final uncertainty in either the glider or the SOOP methane concentration measurements?

L314-315 - The use of “spatio-temporal correlation” in the sentence is unclear. Did you look at computing cross-correlation as a function of lags in space and time between model and observations? Maybe add details or consider rephrasing.

Main Figures:

Fig 1: c,b,d Please use the same scale for methane concentrations as in the text. If I am correct, the units in the text are nano-moles L⁻¹, while those in the figure are mol m⁻³. Also, is it possible to do this figure without using a log scale, or does it lose the prominent visualization of the “hotspot” of the concentrations near the source site of the CH4 release?

Fig 1: Panel a) - consider adding regional markers/boundaries to make it easier to place the information on the map (e.g. country names) and the MPA key used in Figure 3. Also, there is no scale for bathymetry depth. Consider filled contours to show the same depth levels in distinct

intervals if bathymetry is unimportant to this text, making it easier to distinguish relatively deep from relatively shallow areas.

Fig 2 a - same comment on scales as Fig 1 c,b,d. Using color gaps in the axis distracts the eye and makes the figure seem more relaxed. Consider just showing gaps in the data instead of gaps in the axis.

Fig 2 c - see Fig 1 a) place names help. Also, using bathymetry colors without a bathymetry scale is tricky. Maybe consider leaving it with a white background with contours or using filled contours?

Fig 3 - same as Fig a, use of scales for methane concentrations. I also have trouble distinguishing the relative concentrations with the red amplitude colormap and hatchings. This figure is tough to read in black-white. Consider another colormap choice going in 6 increments (<https://mintpy.readthedocs.io/en/latest/api/colormaps/> like `no_green`) using concentration ranges for the exposure, especially given that the current landmask color is quite close to this colormap as well. Also, using blue bathymetry in the small section of the map not covered by the methane levels is not helpful. Just make that white since bathymetry is not essential in this context. Also, please add place names to the relatively large map to make it easy to find where those MPAs are.

Fig 3. Panel 2 with MPAs and concentrations: See the comment above for the color scheme. It might be good to have a key on which MPA is where on the map in Fig 1a)

Supplement Figures:

Fig A1: Glider CTD - O2 data - I would show the gap as a gap not interpolated in between or is this just a result of the `pcolormesh` plotting function?

Fig C3: The figure is quite busy due to the gaps with the beige color. I would just make it a continuous axis with gaps in the data. Also, there are so many data sets on this axis; break it into two subpanels. That might make it easier to visualize.

Fig D5: Consider adjusting the color similar to the recommendation in Figure 3.

Reviewer #1 (Remarks on code availability):

Most of the code to reproduce the figures and research was shared. The only part of the code potentially missing is the code related to the lag corrections of the glider methane sensor data and those related to processing the SOOP vessel-based methane measurements.

Reviewer #2 (Remarks to the Author):

This study utilized continuous high resolution in-situ methane observations from a glider and a ship of opportunity (SOOP) traversing modeled the quantity of dissolved CH₄ over time. However, as a

scientific paper, it lacks a lot of necessary explanations for methods, experiments, and results, which somewhat reduces the readability of the article. Here are my comments on this article:

1. The title is "Nord Stream methane leaks spread across 14% of Baltic waters", but it is not clear from the text how the 14% was obtained. In the conclusion, "..., identifying 23 MPAs that experienced significantly heightened levels of methane." How were the 23 MPAs calculated? What is its connection to the 14% in the title?

2. The article describes the observations in detail, but the description of the model and the calculation process is too simple. Please add an overview of the model and existing applications in the introduction, and add the core algorithms or formulas of the model in section 2.3.

3. How is the total dissolved mass of 10.9 kt given and 10% uncertainty calculated? Why is the dissolved mass range from 9.5 to 14.4 kt?

4. In ChemicalDrift model, "The implementation assumes a negligible methane concentration in the air for simplicity", but the reality is that peak methane concentrations were detected at several in-situ sites. Besides, in the article, the first reason given by Line 176 is that "the initial methane concentrations are more than 3 orders of magnitude higher than a typical background concentration of methane in equilibrium with the atmosphere", which seems to emphasize that the initial methane concentrations are important, so why is this a reason for neglecting methane concentrations in the air?

Minor comments:

1. Lines 44 and 77, 121, 133, 137 and 218, please add spacecraft between numbers and units, e.g., "1 km", "1 Hz".

2. What are the EW and WE-transects? Please give some necessary explanations for the first appearance of the abbreviations.

3. Lines 189-191, is there any reference that confirms that these particle numbers are sufficient to provide robust statistics?

4. What is the gray bar in Figure 2 and C3? Please modify its transparency to avoid obscuring other values.

5. Please add the source of the data for the oceanographic condition near line 241, the full name of SAMBA, data URL (or ref.) in Appendix A.

6. Why is the vertical coordinate of Figure 1c negative but 1d positive?

7. Please check that the names and formats of the axes of the graphs are correct and harmonized.

8. Please check the reference format.

RESPONSE TO REVIEWER COMMENTS FROM REVIEWER #1:

Review of “Nord Stream methane leaks spread across 14 of Baltic Waters”

This study outlines the measurements and mapping of the Baltic Sea surface and subsurface signature elevated CH₄ levels caused by the Nord Stream 2 natural gas pipeline leaks. The authors describe the curation and processing of autonomous glider and vessel-based sea-surface methane concentration measurements with agreement between platforms. These measurements and a lagrangian tracer analysis tool (ChemDrift) fed by a regional ocean forecast system are used to predict the subsurface spread of the methane retained in the water column from the leak sites. Their spread and concentrations in several Baltic Sea Marine Protected Areas (MPAs) are discussed in the text. The authors arrive at an estimate of ~10.9kt dissolved methane from the leak, or roughly 2.5% of the total emitted methane that escaped to the atmosphere (445 kt). The dissolved methane was predominantly stored in the intermediate subsurface layer above a local halocline and below the seasonal mixed layer, allowing it to spread through currents. The authors report increased dissolved CH₄ levels in 23 MPAs by factors ranging from 2 to 100 depending on the distance to the leak site. Some dissolved methane was later entrained into the winter-time mixed layer and outgassed to the atmosphere. Overall, the primary societal impact associated with increased dissolved methane levels in the Baltic from the leak appears to be the additional contribution to Europe’s GHG emissions. The research demonstrates the value of rapid marine environmental assessments in a unique socio-political and economic landscape and physically dynamic environment. The benefits of underwater gliders and other marine observing infrastructure to society are evident.

The paper is clear, well-written, and of broad interest to many disciplines and the general public. The information presented supports the overall findings as presented by the authors. The methodology is clear, but some passages in the text could be improved for clarity. The conclusions arrived at in the study could be strengthened by comparing the results to those anticipated in the initial impact assessment for the Nord Stream 2 project. The long-term impact of the leak on the marine environment near and away from the leak site is also unclear. More context should be provided on how this event impacts the surrounding marine ecosystem. Additional specific comments/suggestions are included below. I think the paper should be accepted with minor revisions.

Thank you for an enjoyable paper to read and review.

Thank you for reviewing our manuscript. We appreciate the clear and concise ideas for improvement of our manuscript and the time and effort you spent for providing the feedback. We have included most of your suggestions, or at some places addressed your concerns in alternative ways. The changes we included in the manuscript are highlighted in blue for new content and red for removed contents (see diff.pdf ,colours not applicable for revised figures and references). All page and line numbers refer to the revised manuscript **with tracked changes**.

General comments:

One aspect I think should be included is how this event compares with the original impact assessment report done by Nord Stream AG, “Chapter 9: Assessment of risks related to unplanned events”. The report mentions the worst-case release of 148kt during a pipeline leak. That is ~3.7 times lower than the worst case that occurred (number in the text 445 +/- 25 kt). The impact assessment also does not discuss any marine impact due to the low solubility of methane. Given

your findings, how would you characterize this report? It would be good to include this original assessment in the introduction and compare the expectations with what happened so that in the future, this type of knowledge could be integrated into future projects of this nature. A few sentences in the introduction/discussion/conclusion section and abstract would enhance this paper's findings.

Thank you for the suggestion. The Nord Stream report mentions the possibility of one ruptured pipeline, which would amount to the release of approximately 148 kt gas. This number is about three times lower than the estimation by Harris et al. (445 kt), because three pipelines were sabotaged simultaneously. I could not reproduce/understand the "3.7 times lower", maybe there is a mistake?

We added a couple of lines about the report in the manuscript on page 2, lines 59-68.

The authors note that the main environmental impact of the pipeline leak will be elevated GHG emissions. What is the broader marine impact of this event, if any? Some additional comments in the conclusion/introduction could relate this question to the findings.

The broader marine impact is not fully understood yet, but we hope the mapping of dissolved methane concentrations give in this study can be of help to scientists researching this. Given the expertise of some of our authors in the field, we included some careful predictions and an outlook of ongoing research to address this relevant question:

Additions to the manuscript on page 11, lines 377-405.

Specific comments:

Correction of the glider CH₄ measurements: The corrections are complicated with three distinct dependencies ($\tau_{\text{ch}_4} + \tau_{\text{temperature}} + \tau_{\text{oxy}}$). I was looking for this correction process online in the GitHub link to the paper review, and I could not find this information. I suggest sharing a code example with data to benefit others in the research community who may want to use this system on either gliders or floats. It appears to be a promising solution.

Thank you, we are happy to share our approach (complete processing code) in our code-repository. It can be found at https://github.com/MartinMohrmann/Mohrmann_et_al_NS_methane_baltic/blob/main/LagCorrection.ipynb

Glider CH₄ sensor integration: It was not mentioned if the CH₄ sensor was set up with a pumped flow head to increase gas flow through the membrane.

The sensor does not have a pumped flow head. We added to the manuscript, page 3, lines 93-96:

The Franatech METS sensor was mounted on the outside of the glider on all configurations (both pumped and unpumped CTD). Flow past the sensor was ensured by the glider's motion through water (approx. 35-40 cm s⁻¹).

L83-84 - Consider mentioning how that calibration was done in the appendix.

The calibration was done by the manufacturer Franatech, so we won't go into much detail here. However, we added a brief explanation to the manuscript, page 3, lines 100-103:

Multi-point temperature and methane calibrations for the 10-1000 nM range were performed by the manufacturer before and after the mission. An additional calibration to determine oxygen sensitivity was performed after the mission.

L86-87 - sentence unclear/ hard to understand.

We simplified the structure of the sentence and added some additional information. You can find the changes on page 3, line 104-109.

L110 - How well did the data agree with each other? What was the final uncertainty in either the glider or the SOOP methane concentration measurements?

Our glider transect does not cross the shipping line, because that would imply high risk of collision with a ship and instrument loss. Thus, we do not have a direct comparison between the SOOP data and our glider observations.

The accuracy of the SOOP system itself is mentioned in the methods section:

The overall accuracy of the methane measurements is usually better than 1 %, though the high concentrations encountered in the first days after the Nord Stream accident fall out of the calibration range.

Similar for our glider sensor, which was calibrated for best sensitivity in a range from 10nM to 1000nM:

“Note that measurements above the 1000 nM upper limit of the methane sensor’s calibration are potentially less accurate.”

The uncertainty due to slow response time of the sensor can increase the uncertainty, especially in situations with strong concentration gradients. This, together with model inaccuracies could be reasons that we get some model/observation mismatch in our linear regression plots in Fig. B2. We implemented advanced time-lag correction algorithms, which have been added to the shared codes repository. The sensor uncertainty of the METS methane sensor on the glider is not specified by the manufacturer unfortunately, but it likely depends on multiple environmental parameters and their gradients. An evaluation of these uncertainties was not in scope of this paper.

However, an indirect comparison of the two different measurements is done in the paper. The amount of dissolved methane in the numerical model was tuned with the Glider data, and then the model data was compared with a linear regression to the SOOP data. This indirect comparison indicated a difference of 20%, see Figure B2. This difference was considered as one component of our uncertainty analysis.

L314-315 - The use of “spatio-temporal correlation” in the sentence is unclear. Did you look at computing cross-correlation as a function of lags in space and time between model and observations? Maybe add details or consider rephrasing.

We reformulated the sentence to make it clearer. You can find the changes on page 10, lines 342-345.

Main Figures:

Fig 1: c,b,d Please use the same scale for methane concentrations as in the text. If I am correct, the units in the text are nano-moles L-1, while those in the figure are mol m-3.

Answer: We changed the the units throughout the paper to nM, which equals nmol l^{-1} , to be more consistent.

Also, is it possible to do this figure without using a log scale, or does it lose the prominent visualization of the “hotspot” of the concentrations near the source site of the CH₄ release?

Since the concentrations decrease quickly and stretch over 3 orders of magnitude, the visualisation on a logarithmic scale reveals much more details compared to a linear scale. For comparison and to support our argument, we include a version of Figure 1 on a linear scale below.

We experimented with different colorscale boundaries, but couldn't arrive at a satisfactory visualisation without a logarithmic scale. However, we included many other small Figure improvements that you suggested, see next comment.

Fig 1: Panel a) - consider adding regional markers/boundaries to make it easier to place the information on the map (e.g. country names) and the MPA key used in Figure 3. Also, there is no scale for bathymetry depth. Consider filled contours to show the same depth levels in distinct intervals if bathymetry is unimportant to this text, making it easier to distinguish relatively deep from relatively shallow areas.

We added country names and outlined MPA regions. Adding MPA keys is a great idea, but Figure 1a felt already dense and crowded, so we added the MPA keys to Figure 3 instead. This figure features a larger map and relevant methane concentrations. We simplified the bathymetry to two levels, with a threshold value of 50 m depth. Please see the updated Figure 1 in the manuscript.

Fig 2 a - same comment on scales as Fig 1 c,b,d. Using color gaps in the axis distracts the eye and makes the figure seem more relaxed. Consider just showing gaps in the data instead of gaps in the axis.

We adjusted the scales to nM and removed the grey vertical spans. They vertical spans did not actually mask anything, but just “filled” gaps in our observations. I agree it looks better to just show the data with gaps. In mid-november, we coincidentally had a gap in both SOOP and Glider missions for unrelated reasons. Please check the updated Figure 2 in the manuscript.

Fig 2 c - see Fig 1 a) place names help. Also, using bathymetry colors without a bathymetry scale is tricky. Maybe consider leaving it with a white background with contours or using filled contours?

Thank you for the suggestions. We added place names and removed the bathymetry information. We tried a white background, but the contrast to the light red colours was too low. Instead, we went with a blue background for the sea which also makes it possible to see how far the methane spread in low concentrations.

Fig 3 - same as Fig a, use of scales for methane concentrations. I also have trouble distinguishing the relative concentrations with the red amplitude colormap and hatchings. This figure is tough to read in black-white. Consider another colormap choice going in 6 increments (<https://mintpy.readthedocs.io/en/latest/api/colormaps/> like `no_green`) using concentration ranges for the exposure, especially given that the current landmask color is quite close to this colormap as well. Also, using blue bathymetry in the small section of the map not covered by the methane levels is not helpful. Just make that white since bathymetry is not essential in this context. Also, please add place names to the relatively large map to make it easy to find where those MPAs are.

Fig 3. Panel 2 with MPAs and concentrations: See the comment above for the color scheme. It might be good to have a key on which MPA is where on the map in Fig 1a)

- We changed the colormaps in Figure 3 to 6 distinct steps. The 6 different shades are easily distinguishable, so we preferred to keep 6 different shades of the red colour instead of using different colours, which could be a bit confusing to read in our opinion.
- Scales changed to nM
- colormap changed to a 6 increments colormap
- bathymetry contours removed (except for 2 step deep/shallow differentiation)
- Country names and MPA keys added

Supplement Figures:

Fig A1: Glider CTD - O2 data - I would show the gap as a gap not interpolated in between or is this just a result of the `pcolormesh` plotting function?

The interpolation was indeed a result of using the python `matplotlib pcolormesh` function. However, we removed (technically covered with a white patch) the interpolation for the data-gap of several days. Please check Figure A1 in the manuscript.

Fig C3: The figure is quite busy due to the gaps with the beige colour. I would just make it a continuous axis with gaps in the data. Also, there are so many data sets on this axis; break it into two subpanels. That might make it easier to visualize.

As suggested for earlier figures, we just leave out the visual bars to mark data gaps. The Figure shows model data subsampled at the gliders' location, so it shows gaps for periods when there was no glider in the water. The previous visualisation of the observation data is also gone. It could have been included into a sub-panel, yes, but since that sub-panel would then be nearly identical to Figure 2a, we decided to simply leave it out. Please check Figure C3 in the manuscript.

Fig D5: Consider adjusting the colour similar to the recommendation in Figure 3.

We introduced the same new colour scale, please check Figure D5 in the manuscript.

Reviewer #1 (Remarks on code availability):

Most of the code to reproduce the figures and research was shared. The only part of the code potentially missing is the code related to the lag corrections of the glider methane sensor data and those related to processing the SOOP vessel-based methane measurements.

The code to process the glider methane sensor data is shared now in our code repository (https://github.com/MartinMohrmann/Mohrmann_et_al_NS_methane_baltic/blob/main/LagCorrection.ipynb).

For the SOOP measurements, calculation of the methane concentration data has been described in (14, 15) as described in the manuscript. The – in this case essential – time lapse correction, has been described in (15, Appendix), based on the algorithm introduced (for O₂-sensors) in (17), again, given in the manuscript). The derivation of the time lapse time scale, tau, from a correlation with the water flow briefly mentioned in the manuscript was necessary because of a gradual decrease in water flow rates. Due to the importance of the data, it had been decided not to shut down the system for repair work on the water supply system, as this would have unavoidably resulted in a loss of at least 2 weeks of data. In that regard, this tailored post-processing was based on correlation and analysis and experience and could not be transferred on a generally usable code to be posted on e.g. GitHub.

RESPONSE TO REVIEWER COMMENTS FROM REVIEWER #2:

This study utilized continuous high resolution in-situ methane observations from a glider and a ship of opportunity (SOOP) traversing modeled the quantity of dissolved CH₄ over time. However, as a scientific paper, it lacks a lot of necessary explanations for methods, experiments, and results, which somewhat reduces the readability of the article.

Thank you for the review of our manuscript. Your constructive feedback helped us to improve our manuscript and we believe we explained the critical sections more clearly now. The changes we included in the manuscript are highlighted in blue for new content and red for removed contents (see diff.pdf ,colours not applicable for revised figures and references). All page and line numbers refer to the revised manuscript **with tracked changes**.

Reviewer #2 Comments:

1. The title is "Nord Stream methane leaks spread across 14% of Baltic waters", but it is not clear from the text how the 14% was obtained. In the conclusion, "..., identifying 23 MPAs that experienced significantly heightened levels of methane." How were the 23 MPAs calculated? What is its connection to the 14% in the title?

Thank you, this was indeed not described well. We added some additional clarification to the methods section. See page 7, lines 209-220.

Moreover, in the results section (page 11, lines 375-377), we now explicitly mention the threshold of our 14% calculation:

[...]The model shows that 14% of the Baltic Sea area experienced concentrations five times above background levels (> 50 nM).[...]

In this context, the 14% of the Baltic Sea area percent of the complete area of the Baltic Sea, and these 14% area experienced dissolved methane concentrations 5 times greater than average natural levels (>50nmol l⁻¹). We introduced an average of "natural levels" in the text to make the values in the abstract more accessible for readers that are not familiar with common dissolved methane concentrations in the sea.

2. The article describes the observations in detail, but the description of the model and the calculation process is too simple. Please add an overview of the model and existing applications in the introduction, and add the core algorithms or formulas of the model in section 2.3.

The OpenDrift libraries are an established toolbox in the field that were used in more than 60 publications (<https://opendrift.github.io/references.html>). In the manuscript, we added a brief introduction of the model in the Introduction section. See page 3, line 77-80.

Moreover, we added model equations and a table with the key algorithms and parameters in the methods section 2.3. (page 6, lines 195-220)

3. How is the total dissolved mass of 10.9 kt given and 10% uncertainty calculated? Why is the dissolved mass range from 9.5 to 14.4 kt?

The mass of dissolved methane was estimated by iterative linear regressions between the modeled versus observed concentrations. The variable that was tuned in the process was the fraction X of the methane escaping the pipelines that went into solution. The time dependency of the escaping gas was defined by CATHARE/PBREAK engineering estimates of gas escaping the pipelines (presented in joint-publication).

In practise the process looked like this: First we ran the model with an educated guess about the amount of dissolved methane, based on observations and rough estimations and then we compare the output of the model with our glider observations in a linear regression plot (Figure B2). We found the slope of the regression plot to be 0.70, so we started a second model run with $1/0.70=1.4$ times more methane and checked the linear regression again. Due to non-linearities, we had to repeat this process a couple of iterations, until we reached a linear regression slope of 1.00 between model and observations. Due to the low system requirements and good performance of the Lagrangian model, this process wasn't too cumbersome.

The mentioned 10% uncertainty is from the ordinary least squares regression used to tune the initial model amount of dissolved methane to the observations (Figure B2). The linear regression engine returned that one standard deviation of the slope is $\sigma=0.05$. We chose to use a 2σ environment for our uncertainty calculation, resulting in a slope of 1.00 ± 0.1 , or 10% uncertainty.

However, the uncertainty from the slope of the linear regression is just one component in the uncertainty analysis. The total (combined) uncertainty for the amount of dissolved methane consists of uncertainty estimations from multiple components:

1. Two different numerical engineering models of the Nord Stream gas emissions were used (PBREAK and CATHARE). We implemented both model data as boundary conditions for our model, running the model twice to compare.
2. We used different observation data as base for our ordinary least squares regression. We used (a) first two days of Glider observations, (b) complete time series (3 months) of glider observations, (c) SOOP surface observations. The resulting slopes give an indication how much the amount of dissolved methane differ between model (reference scenario) and each observation dataset.
3. The PREAK and CATHARE models submitted three different scenarios (lower, reference, upper).

In summary, we added the uncertainties caused by each individual component to a total uncertainty. We added some additional explanations in appendix B, page 21 to make our method more transparent.

4. In ChemicalDrift model, "The implementation assumes a negligible methane concentration in the air for simplicity", but the reality is that peak methane concentrations were detected at several in-situ sites. Besides, in the article, the first reason given by Line 176 is that "the initial methane concentrations are more than 3 orders of magnitude higher than a typical background concentration of methane in equilibrium with the atmosphere", which seems to emphasize that the initial methane concentrations are important, so why is this a reason for neglecting methane concentrations in the air?

Atmospheric methane concentrations, even shortly after the leaks would only have an insignificant effect on the concentrations of dissolved methane in the seawater in the vicinity of the leak sites. We would like to support this argument with a scale analysis: A typical atmospheric background methane concentration is about 1900ppb. Atmospheric methane concentrations after the Nord Stream leaks were observed/ modeled by

1. Reum et al. (In Review for Nature Communications) measured in situ atmospheric methane concentrations (5 Oct) of up to ~2380 ppb.
2. CAMS simulations (<https://atmosphere.copernicus.eu/cams-simulates-methane-emissions-nord-stream-pipelines-leaks>) indicate a maximum of about 2800 ppb.
3. GHG sat measurements indicate an enhancement above background values of up to 1000 ppb.

We here assume, as an example, the upper boundary of atmospheric methane concentrations of 2900 ppb (highest value of the above). An equilibrium between an atmosphere with 2900 ppb methane and seawater (e.g. at 14°C) results in concentrations of dissolved methane of well below 10 nmol l⁻¹. Equations to compute this can be found at <https://doi.org/10.1021/acs.est.5b01261> or in our released source code (methane_utils.py). The equilibrium solubility is dependent on the temperature and salinity, but for our parameter space the concentrations are negligible because the resulting contributions would be below the threshold that we call “natural methane concentrations”, and 2-3 orders magnitude smaller than the observed dissolved methane concentrations for the month after the leaks. Thus after the pipeline ruptures, the water was a relevant source of atmospheric methane, but the atmospheric methane was not a large source of dissolved methane in the seawater. Instead, the dissolved methane was caused earlier, while the methane was bubbling from the depth through the water column towards the surface.

Minor comments:

1. Lines 44 and 77,121,133,137 and 218, please add spacecraft between numbers and units, e.g., "1 km", "1 Hz".

Thanks, we checked all units and added spacecrafts where necessary.

2. What are the EW and WE-transects? Please give some necessary explanations for the first appearance of the abbreviations.

Changed to East-West and West-East transects. Given the reference to the map with the glider observation line, we believe this is clarified enough.

3. Lines 189-191, is there any reference that confirms that these particle numbers are sufficient to provide robust statistics?

We couldn't find a citable paper that relates Lagrangian particle numbers to statistical model properties. Thus, we removed the statement that the used numbers provide robust statistics from the text.

However, we reason that we consider numbers of $n > 50$ particles in a grid cell as sufficient was the following: The error of discretization, e.g. if a certain Lagrangian particle is located (binned) in one grid-cell or a neighbouring grid-cell is in the order of n , and smaller than the uncertainties estimated from other sources (e.g. linear regression, engineering conditions of pipeline methane release etc...)

4. What is the gray bar in Figure 2 and C3? Please modify its transparency to avoid obscuring other values.

We removed the bar altogether. It was just a vertical span that marked the absence of observations. Nothing was obscured behind, but it looks better without.

5. Please add the source of the data for the oceanographic condition near line 241, the full name of

SAMBA, data URL (or ref.) in Appendix A.

We included this information in Section 6, Data availability:

[...] Furthermore, we used datasets SEA077_M11, SEA069_M11, SEA_077_M13, SEA045_M69, SEA077_M15, SEA045_M71, SEA077_M17 and SEA045_M73 to describe regional oceanographic conditions in Section 3 and Figure A1.

6. Why is the vertical coordinate of Figure 1c negative but 1d positive?

We unified this to have “positive depth downwards” coordinates in all panels and plots.

7. Please check that the names and formats of the axes of the graphs are correct and harmonized.

Yes, this was confusing before indeed! We harmonized all units of methane concentration in the text and figures to the units nM (nmol l^{-1}).

8. Please check the reference format.

We corrected many small errors in the references.

REVIEWERS' COMMENTS

Reviewer #1 (Remarks to the Author):

Dear Dr. Mohrmann and Colleagues,

Thank you for submitting the revised manuscript and for your detailed comments. I believe this version successfully addresses the majority of my initial concerns and suggestions. I am looking forward to seeing this paper published. Below, I have listed a few minor comments that I believe should be considered for the final publicly available version of the manuscript.

I wish you the best of luck with the continuation of the review process.

Sincerely,

Nicolai von Oppeln-Bronikowski

Memorial University
St. John's, Canada

L17: In the Abstract please, check the total of suspected methane. Should it be 410-480 kt (445+/-35 kt)?

L46: 445+/-35 kt - make sure the number and updated uncertainty is appropriately revised everywhere in the text.

L59-63: In my opinion: I would integrate and move these two sentences to the first paragraph (L37-47) where you mention the total escaped mass of methane. After I would add something like this: "In the environmental assessment report commissioned by the pipeline operator (Nord Stream AG), the mass of methane with possibility of escape is 148kt from a single pipeline rupture [5], and is in agreement with [1] for the case of a single pipeline rupture. However, the possibility of multiple pipeline ruptures, in this case three, at the same time was neglected and thus the actual total methane release (410-480 kt) was about three times greater than the worst case estimate (148 kt) included in the report."

L63-67: "The probability of a pipeline failure" I would move these two sentences to the discussion into the conclusions. Also consider changing the use of "after" twice in one sentence. Perhaps reword like this: "The rupture of the Nord Stream pipelines in 2022 and Baltconnector pipeline between Finland and Estonia only XX years after commissioning [6], requires that the impact assessments for these type projects be adapted to the current geopolitical context.

L396: You should add a source for the Russian gas composition.

Reviewer #2 (Remarks to the Author):

Overall, the authors' responses to all suggestions were comprehensive and detailed, making valuable improvements to the manuscript. I only have a few minor suggestions now.

1. Lines 180 and 181, please use the mathematical symbol " \times " instead of the letter "x".
2. Line 629, "... the slope ibn an ordinary least squares fit...", What does "ibn" mean here?
3. Please replace "Suppl." and "Supplementary" with "Appendix" in the main text.
4. Figure 1, Lack of caption for the horizontal coordinate of figure 1c. change the color of thermocline in Figure 1d, as the colour is too close to the dark red part of the background.
5. Figure 2, the labels in this figure needs to be changed to "Methane conc. [nM]"
6. Figure 3, What do the numbers on the Figure a mean?
7. References, the format of the references does not seem to match the requirements of the journal. Formatting between references also remains inconsistent, with some missing page numbers (e.g., [2], [9], [10], [17], [28]) and some having two doi (e.g., [14], [24]). In addition, papers in the submission (i.e., [1], [31]) may not be placed in the reference list. Reference [33] is available in an officially published version (<https://doi.org/10.5194/acp-24-4675-2024>).
8. Fig. C4, lack of label a and b on the figure.

Reviewer #2 (Remarks on code availability):

The author seems to have provided the complete code and the Readme file, but I didn't try to run the code.

Response to reviewer 1:

L17: In the Abstract please, check the total of suspected methane. Should it be 410-480 kt (445+/-35 kt)?

We can see how the numbers appear to contradicting each other at first read. In the abstract we specified to the amount of methane that **escaped from the pipelines** according to the CATHARE pipeline rupture modelling presented in [1], while in the introduction we specified the amount of methane that **reached the surface**. This is admittedly confusing without having read the companion manuscript [1] first.

We realized that by quoting the amount of methane that reaches the atmosphere, we implicitly anticipate the result of our own study, which was used in [1] alongside alternative studies to derive the amount of dissolved methane remaining in the depth of the Baltic Sea for a prolonged time or being biodegraded (oxidized).

To make this less confusing, we now specify instead the amount of methane escaping the pipelines in the depth, from both used pipeline rupture models (PBREAK, CATHARE), including their uncertainty estimates. These are the numbers that are actually used in our model. We remove the explicit statement about the amount of methane reaching the surface and leave a reference to [1] instead, where our and many more Nord Stream leak related studies are briefly presented and put into context.

Relevant changes on **page 2, 42-44**.

L46: 445+/-35 kt - make sure the number and updated uncertainty is appropriately revised everywhere in the text.

See previous response.

L59-63: In my opinion: I would integrate and move these two sentences to the first paragraph (L37-47) where you mention the total escaped mass of methane. After I would add something like this:

“In the environmental assessment report commissioned by the pipeline operator (Nord Stream AG), the mass of methane with possibility of escape is 148kt from a single pipeline rupture [5], and is in agreement with [1] for the case of a single pipeline rupture. However, the possibility of multiple pipeline ruptures, in this case three, at the same time was neglected and thus the actual total methane release (410-480 kt) was about three times greater than the worst case estimate (148 kt) included in the report.”

Thank you, we changed this following your advice (with minor changes). Changes on page 1, **45-55**.

line

L63-67: “The probability of a pipeline failure” I would move these two sentences to the discussion into the conclusions. Also consider changing the use of “after” twice in one sentence. Perhaps reword like this: “The rupture of the Nord Stream pipelines in 2022 and Baltconnector pipeline between Finland and Estonia only XX years after commissioning [6], requires that the impact assessments for these type projects be adapted to the current geopolitical context.

Thank you for the suggestions. We moved **L63-67** to the discussion and reworded it following your suggestion. It can now be found on page 12, **line 300-305**.

L396: You should add a source for the Russian gas composition.

Added.

Reference used in this response:

[1] Harris, S.J., et al.: Methane emissions from the Nord Stream subsea pipeline leaks. Nature (Article submitted alongside this manuscript, tracking number: 2023-11-20498) (2024)

Thank you for the helpful comments and the review of our manuscript.

Response to reviewer 2:

Overall, the authors' responses to all suggestions were comprehensive and detailed, making valuable improvements to the manuscript. I only have a few minor suggestions now.

1. Lines 180 and 181, please use the mathematical symbol " \times " instead of the letter "x".

Corrected.

2. Line 629, "... the slope ibn an ordinary least squares fit...", What does "ibn" mean here?

Sorry, that was a typo. It is now corrected to "*the slope of the ordinary least squares fit*"

3. Please replace "Suppl." and "Supplementary" with "Appendix" in the main text.

The editor recommended us to have an extra document with supplementary materials instead of an appendix. Since we copied the appendix into an extra supplementary document during this revisoin, we keep the term "supplementary".

4. Figure 1, Lack of caption for the horizontal coordinate of figure 1c. change the color of thermocline in Figure 1d, as the colour is too close to the dark red part of the background.

Added caption, line color changed from red to black. Thank you for the helpful comments.

5. Figure 2, the labels in this figure needs to be changed to "Methane conc. [nM]"

Thanks, added abbreviation dot in 'conc.'

6. Figure 3, What do the numbers on the Figure a mean?

The numbers are references to the number of the respective MPA in Appendix Figure D5. This important information was indeed missing in the Figure caption, thank you for noticing this.

7. References, the format of the references does not seem to match the requirements of the journal.

Formatting between references also remains inconsistent, with some missing page numbers (e.g., [2], [9], [10], [17], [28]) and some having two doi (e.g., [14], [24]). In addition, papers in the submission (i.e., [1], [31]) may not be placed in the reference list. Reference [33] is available in an officially published version (<https://doi.org/10.5194/acp-24-4675-2024>).

- We added page numbers where available. Some journals have continuous online publishing and do not add page numbers to their articles any longer. Some use Article Numbers instead of page numbers.
- We removed the double DOIs/URLs for references [13], [22], [26] and [27].
- We aim for a joint-publication with the authors of [1], [31] and have communicated this clearly to the Editors. This means upon publication of our manuscript the other manuscripts will also become available. In the Nature Comms. reference guide it says: "*Only papers that have been published or accepted by a named publication or recognized preprint server should be in the numbered list.*" I believe it is easiest to leave the details of how these references shall be handled to the editor.
- We replaced reference [33] with the published version.

8. Fig. C4, lack of label a and b on the figure.

Added, thanks.

Thank you for the review of our manuscript.